# Effects of selenium nanoparticle on the growth performance and nutritional quality in Nile Tilapia, Oreochromis niloticus

Fasil Dawit Moges[1], Hamida Hamdi[2]*, Amal Al-Barty[2], Abeer Abu Zaid[3], Manisha Sundaray[1], S. K. S. Parashar[4], Abebe Getahun Gubale[5], Biswadeep Das[1]*

**1** School of Biotechnology, Kalinga Institute of Industrial Technology, KIIT Deemed to Be University, Bhubaneswar, Odisha, India, **2** Department of Biology, College of Science, Taif University, Taif, Saudi Arabia, **3** Department of Biology, Alkhormah University College, Taif University, Taif, Saudi Arabia, **4** School of Applied Science, Kalinga Institute of Industrial Technology, KIIT Deemed to Be University, Bhubaneswar, Odisha, India, **5** Department of Zoological Sciences, Addis Ababa University, Addis Ababa, Ethiopia

* biswadeep.das@kiitbiotech.ac.in (BD); Shimaa76sl@tu.edu.sa (HH)

**Data Availability Statement:** All relevant data are within the paper.

## Abstract

Selenium is an important micronutrient that has antioxidant, growth potential, and reproduction enhancement abilities in various organisms. The aquaculture industry is a significant contributor towards meeting the dietary requirements of a majority of the global population, which further warrants developing novel approaches for enhancing the production of dietary fish. This study was performed to assess the growth performance of Nile tilapia (*Oreochromis niloticus*) fingerlings (1 gm in average weight and 2.75 cm in average length) upon nano-selenium (Se-Nps) supplementation. Nanoselenium was synthesized using high-energy ball milling (HEBM) using a 10-hour dry milling technique at 10:1 ball-to-powder ratio (BPR), size characterized by XRD and TEM, followed by mixing with basal feed in desired concentrations (0.5 mg/kg, 1 mg/kg, and 2 mg/kg) and administration to Nile tilapia fingerlings for 30 days, followed by the evaluation of growth performance parameters, fatty acid profile analysis using GC-MS, and nutritional quality index (NQI): [Thrombogenicity Index (IT), Atherogenicity Index (IA), n-3/n-6, n-6/n-3)]. Nile tilapia supplemented with 1 mg/kg Se-Nps showed improved growth performance (RGR: 1576.04%, SGR: 4.70%, and FCR: 1.91), demonstrated by higher survivability (> 95%), isometric growth (coefficient of allometry, b = 2.81), and higher weight gain compared to control (RGR: 680.41%, SGR: 3.42%, and FCR: 1.31), 0.5 mg/kg Se-Nps (RGR: 770.83%, SGR: 3.61%, and FCR: 1.18) and 2 mg/kg Se-Nps (RGR: 383.67%, SGR: 2.63%, and FCR: 1.22). The average length-weight relationship assessed as the condition factor (K) was highest in the 1 mg/kg Se-Nps group compared to others (p < 0.05). GC-MS analysis revealed that Nile tilapia supplemented with 1 mg/kg Se-Nps showed better meat quality, higher amount of n-3 fatty acids, eicosapentaenoic acid, and docosahexaenoic acid, high PUFA/SAFA ratios (1.35) and n-3/n-6 (0.33) ratios, with low atherogenicity index (0.36) and thrombogenic index (0.44), and relatively low n-6/n-3 fatty acid ratio (3.00) compared to other groups. Overall, Se-Nps supplementation at 1 mg/kg enhanced the growth performance and meat quality in Nile tilapia, and therefore could be a potential growth-promoting micronutrient for aquaculture enhancement.

**Funding:** The author(s) received no specific funding for this work.

**Competing interests:** The authors have declared that no competing interests exist.

## 1. Introduction

The contribution of fish farming to global food requirements is huge owing to the increasing demand for fish as a primary nutritional source in several countries. The global aquaculture fish production reached 82.1 million tons in 2018, approximately 46% of the world's fish production [1]. Nile tilapia (*Oreochromis niloticus*) is one of the commercially important fish, forming the basis of commercial fisheries in many parts of the world. Tilapia is a freshwater fish, which is one of the most widely reared fish across the world. The importance of Nile tilapia could be attributed to its versatile applications, including fast growth, ease of breeding in captivity, high productivity, high fecundity, major food source that caters to most human populations, competitive market price, stress and environment tolerant, and many others [2, 3]. Tilapia culture has emerged as one of the most convenient and highly profitable fish farming methods in over 120 countries [3]. In 2017, tilapia was ranked as the fourth most popular culture fish and included in the top ten aquaculture-producing fish species groups across the world [4]. This trend coupled with the need to supply the world with more quality food has instigated researchers to explore various growth improvement techniques for tilapia culture [5].

The recent advent of nutrition-sensitive aquaculture for the production of diverse variety of sea food for supporting the food requirements of the general public, providing adequate amounts of recommended dietary elements, and being nutritionally and culturally safe for acceptance in the global population is a promising approach that assists in the identification and utilization of specific fish species for meeting the nutrition demand of the global population [6, 7]. In this regard, tilapia is a protein rich fish, and is inexpensive to rear, thereby possessing potential to meet the daily dietary requirement of a large population, including poor and malnourished people. In several underdeveloped and developing countries, tilapia provides a staple protein diet for a large population and is therefore considered an important dietary fish. Given the importance of tilapia as a dietary and commercial fish across the world, information on the nutrition potential and aquaculture productivity of this species are crucial. The quantitative parameters of fish, such as condition factor, length-weight relationship, fecundity, and mortality are important aspects for the assessment of growth performance and the nutrient status in natural environments [5, 6], and which have been applied to assess the effect of environmental alterations on the well-being of fish [6]. Condition factor is an index reflecting interactions between the biotic and abiotic factors with the physiological condition of fish. It has been demonstrated that heavier fish reflects a healthier physiological state [7]. Condition factor has been used as an indicator of health in fishing biology studies such as growth and feeding intensity. It can be used to determine the feeding activity of a species to determine whether it is making good use of its feeding source and obtaining sufficient energy through the diet [7].

Another feature for the assessment of fish quality is the type and variety of lipids and fatty acid composition in fish [8]. Fatty acids (FA) are one of the components of lipids or fats that are present in structures called triglycerides. FAs are either saturated (SFA) or unsaturated (UFA). In the SFA, the alkyl chain does not contain any double bonds, whereas UFAs contain one or more double bonds. Considerable variation occurs between the fat and lipid content of fish within and between species [8–11]. In general, fish is a healthy, easily digestible food, which is rich in UFAs and poly UFAs (PUFAs). The consumption of fish and fish lipids can provide PUFAs, in particular ω-3 PUFAs. PUFAs derived from fish oil and mixed with basal feed have been reported to aid in the growth and productivity of Nile Tilapia [12]. Because PUFAs are generally not produced by fish, supplementation of PUFAs such as linoleic acid (LOA, 18:2n-6) and alpha-linolenic acid (α-LNA, 18:3n-3), or administration of PUFA

generating nano-minerals in the basal diet would assist in enhancing the growth of fish, which would convert the long chain PUFAs to partial long-chain PUFAs, such as docosahexaenoic acid (DHA), eicosapentaenoic acid (EPA), and arachidonic acid (AA) that act as antioxidants and assist in fish metabolism and productivity [13].

Dietary supplementation of micronutrients in the form of nanoparticles as feed supplements has been reported to enhance the productivity and meat quality of fish, e.g., nFe, nSe, nAg, and nZnO [14]. Nanosupplementation of micronutrients offers several advantages, such as greater bioavailability, easy absorption, enhanced utilization, and promoting cellular functions. Selenium is a trace element and has an important role in the development and immune status of organisms [15, 16]. Several researchers have conducted selenium supplementation for enhancing the productivity and growth performance in *Carassius auratus gibelio* [17], *Ctenopharyngodon idellus* [18], improved productivity and growth in stress-resilient fish [19] improved immune status in *Cyprinus carpio* [20] and juvenile *Lates Calcarifer* Block [21]. However, studies on the assessment of Se-Npssupplementation on the growth performance in tilapia are lacking. The present study assesses the growth performance, length-weight relationship, condition factor (FCF), fatty acid analysis and nutrition quality index, and dietary levels of Se-Nps on the fingerlings of *O. niloticus* to improve the fish production and yield level and quality.

## 2. Materials and methods

### 2.1. Synthesis and characterization of Se-NPSNPSSNpsss

Selenium pellets of about 5 mm, > 99.99% trace metal basis were purchased from Sigma Aldrich, India. The synthesis of the dietary Se was performed in the Nano Lab., School of Applied Science, KIIT University, India using the High Energy Ball Milling (HEBM) through dry milling process according to our previous method [22]. Characterization of the synthesized Se-Nps was performed using XRD and TEM analysis, in addition to assessing the stability of the particle size distribution of the Se-Nps using a zeta sizer.

### 2.2. Tilapia rearing set up and breeding

All the methods were approved and carried out under relevant guidelines and regulations of the Institutional Animal Ethical Committee (IAEC) of Addis Ababa University (AAU), Ethiopia. Eight glass aquariums with 70*40*35 cm dimensions were set at the Green House-Based Research Recirculating Aquarium system facility (RAS), AAU based on the FAO RAS guideline [23]. Four filtering mechanisms were employed: mechanical (physical), biological, carbon filter, and UV light-based filtration were designed and set into three 250 L plastic barrels. A 2000 L plastic barrel was used for de-chlorinated water supply for the recirculating system and also as water backup. The dechlorinated and filtered water was pumped into the recirculating aquarium system using an electric water pump (0.5 HP). The utilized water flows back as overflow beyond the outlet level into the filtering mechanism starting the physical filtration using cotton pads. Unused feed and any physically visible wastes were siphoned using water pumps after an hour of every feeding. 2/3 of the overall recirculating water system was drained and the aquariums are cleaned every alternating day. Water quality parameters: temperature, pH, dissolved oxygen, and conductivity were maintained at optimum conditions in a semi-automated water circulatory Aquaneeering system. The parameters were measured for two to three days per week in the first experimental month, until the RAS attained stability, following which the measurement was done twice per month at the end of the experiment. Parent stocks of Nile tilapia (*O. niloticus)*, were collected from the wild (Lake Chamo, Ethiopia). Healthy parents were selected and put into the brooding tank in the hatchery facility within the greenhouse. Brood-stock conditioning and stocking for breeding were done for brooder-parents at

a ratio of 1:5 (Female to Male) for 12 days. At the end of the breeding days, each mother was checked for fertilized eggs. Eggs were collected manually from the mouth of brooding mothers and transferred into a bowl with dechlorinated water by carefully pressing the mouthparts of the mothers. The collected eggs were transferred to the hatchery set-up. The eggs were let in the recirculating hatchery system for a week.

## 2.3. Stocking and rearing experimental fishes

At the end of the week, the hatched fingerlings with exhausted yolk sac were transferred to the nursery pond/tank. Feeding was provided with a fine size of basal feed only after their yolk sac was completely drained. The young fingerlings (2 month-old) of the same cohort with an average body weight of 2.5 gm and 4 cm in length were randomly selected and stocked into the research RAS set-up at a stocking density of 50 fish per aquarium. The fingerlings were acclimatized in the recirculating system for two weeks. Unhealthy fish and dead individuals were removed and the stocking density was reduced up to 40 healthy fish per aquarium. After completing the acclimatization period, the fish were raised for a total of eight weeks.

## 2.4. Feed formulation and nano-Se dietary supplementation

The basal feed was formulated with a calculated proximate composition of 35.06%, 32.54%, 12.00%, 18.00%, 2.00%, and 0.40% that corresponded to fish meal, soya bean, corn, wheat, CMC (carboxymethyl cellulose) sodium salt, and multivitamin, respectively (Table 1).

Dietary Se-Nps was mixed with the basal feed using a fine powder mixer (IKA-EUROSTAR, power control visc) to make maximum homogenous treatments. The prepared control feeds and mixed experimental feed were transformed to very thick dough using distilled water. The dough was then extruded separately for each inclusion into pellets, followed by packing and storage at a cold store. The experimental supplement inclusions were: 0.5, 1.0, and 2.0 mg Se-Nps/kg basal feed. Feeding of the experimental fish was performed using the experimental feed twice per day. After every meal of feeding, the remaining feed and waste pellets were siphoned to maintain a healthy and suitable culture medium. The feed intake were calculated by the formula:

$$FI = 1.4 * BW^{-0.350}.$$

Where, FI: Feed Intake (gm/kg), BW: Average Body Weight of fish in Kg.

In addition, protein efficiency ratio (PER), which is a major parameter to assess tilapia growth was calculated as:

$$PER = FCR * \frac{\%FP}{\%PCS}$$

**Table 1. Composition, protein and energy content of basal feed.**

| No. | Ingredient | | | GE | | CP |
|-----|-----|-----|-----|-----|-----|-----|
| | Names | % | Gram | MJ/Kg | Kcal/Kg | |
| 1 | Fish Meal | 35.06 | 350.60 | 18.56 | 4433 | 60.30 |
| 2 | Soya Bean | 32.54 | 325.40 | 17.52 | 4184 | 46.47 |
| 3 | Wheat | 18.00 | 180.00 | 16.27 | 3886 | 13.90 |
| 4 | Corn | 12.00 | 120.00 | 17.34 | 4142 | 9.48 |
| 5 | CMC | 2.00 | 20.00 | - | - | - |
| 6 | Vitamin Premix | 0.40 | 4.00 | - | - | - |

Where, CMC: Carboxymethyl Cellulose, Binder. GE: Gross Energy in MJ/Kg (Mega Joules per Kilo Gram), Kcal (Kilo Calorie), CP: Crude Protein.

Where PER: Protein efficiency ratio, FCR: Feed conversion ratio, FP: feed protein, which is the percentage protein contribution of the treatment feed, PCS: Protein in culture species, which is the proximate protein content of tilapia Fingerlings.

## 2.5. Length-weight relationship analysis

The length-weight relationship was calculated using the equation: $W = a^*L^b$

Where W is the weight gain in gm for every length L (TL) in cm. "a" and "b" are growth parameters. The wellbeing or the condition of the fish was calculated as FCF (Fulton's Condition Factor):

$$K = \frac{Weight\ (W)}{Length\ (L3)},$$

Where K is the condition factor (FCF), W is average weight (Av.) in gm, and L is length (Av. TL) in cm. To express the physical growth of tilapia, biometry indices were used. These biometry parameters were carried out at the beginning of the experiment every week, and at the end of the experiment. Bodyweight gain (BWG), specific growth rate (SGR), and relative growth rate (RGR) were calculated as:

$$BWG = BWF - BWI, \qquad \qquad \text{Eq 1}$$

Where, BWG is weight gain (gm), BWF is the final weight in gm, and BWI is the initial body weight in gm.

$$SGR = \frac{(ln\ BWF - ln\ BWI)}{T} * 100, \qquad \qquad \text{Eq 2}$$

Where: SGR is specific growth rate and T is total cultured time.

$$RGR\ (\%) = 100 * \frac{BWF - BWI}{IBW}, \qquad \qquad \text{Eq 3}$$

Feed Conversion ratio, which is more related to growth performance, is calculated as

$$FCR = \frac{Total\ Feed\ Provided\ (gm)}{Total\ Weight\ Gain\ (gm)}. \qquad \qquad \text{Eq 4}$$

Where the total feed provided is the amount of feed used to raise live fish till the end of the experimental period, and the total weight gain is the difference in the biomass between the initial and the final weight.

The initial total weight for each treatment (WT) and length (TL) for the subsamples was measured on day one, followed by every 7th day of the week. At the end of the experiment (the 8th-week), growth parameter measurements and length-weight data were taken for each individual of all treatments. The total length of each fish was measured from the tip of the snout (mouth closed) to the end of the caudal fin using a measuring board. Blood was collected using a combination of two methods; "dorsal aorta puncture" and caudal venous puncture" procedures so that blood samples could be collected from both arteries and veins of freshly cut gills and caudal fins, followed by extraction of whole liver and tissue samples for Se level analysis. Moreover, tissues (muscle) collected before and after the experiments were used for fatty acid profile analysis. Fish and proximate feed analysis were also done separately. Dead fish were removed; if any, from each aquarium and weighed to calculate feed conversion ratio (FCR).

## 2.6. Gas Chromatography-Mass Spectrometry (GC-MS)

Gas Chromatography/Mass Spectrometry, or GC/MS analysis, is an analytical method that combines the features of gas chromatography and mass spectrometry to identify different substances; in this case FAs. GC/MS is used for the quantitative determination of fatty acids in foods and other sample matrices by assessing the conversion of fatty acids into methyl esters (FAME).

Briefly, FA profiling was done by isolating the lipid fraction through Soxhlet extraction using n-hexane for six hours. The lipid portion was then allowed to undergo saponification and derived to FAME (Fatty-acid Methyl Ester) by HCl/Methanol. Individual FAs profiling was done by Gas Chromatography (GC- USA Agilent 7890-B Technology) equipped with a split injector with a flow rate of 100 mL/min. The percentages of the individual FAs were determined using peak areas. Electron Ionization (EI) system was used to separate and quantify each FAME component. The FAMEs were separated using a highly polar HP-88 (USA, Agilent) column (30 m* 0.25 μmm* 0.20 μmm) in the Mass Spectroscopy detector (MSD- Agilent 5977-A). The carrier gas was helium (He) at a linear velocity of 1 ml/min. The oven temperature was 40°C for 2 min and increased by 15°C/min to 130°C, at a rate 4°C/min then to 150°C, at a rate of 3°C/min then to 160°C held for 17 min, at a rate of 5 °C/min, to 205° C and finally at a rate of 15 °C/min to 230° C. Data analysis was performed by determining the percentage area of the FAs and the absolute concentration of specific FAs using external standards and NIST-17 Library (National Institute of Standards and Technology Standard Reference Data Program, Gaithersburg, MD 20899) using Agilent MassHunter (GC/MS, Acquisition: B.07.03.2129 18-Agilent Technologies, Inc.) software.

## 2.7. Nutritional quality indexes of lipids

The nutritional quality of the lipid portion of the samples was determined through the fatty acid compositions. This evaluation was performed using four nutritional quality indicators (indexes) based on fatty acid compositions. The atherogenicity and thrombogenicity indices were calculated according to [24], to assess the biological effects of different fatty acids:

### Atherogenicity Index (IA)

$$IA = \frac{[(C12:0 + (4x\ C14:0) + 16:0)]}{[\sum MUFA + \sum n6 + \sum n3]}$$

IA represents the relationship of the sum of the major saturated fatty acids to that of the major classes of unsaturated, the former being considered pro-atherogenic or pro-thrombotic, and the latter anti-atherogenic or anti-thrombotic, thereby preventing the appearance of micro- and macro- coronary diseases [24].

IA index relates to the content of SFAs with the content of MUFA and PUFAs w-3 and w-6. Generally, the consumption of foods/fish meat with a lower IA can reduce the level of total cholesterol and LDL-C (Low- Density Lipoprotein Cholesterol-"Bad Cholesterol") in human blood.

### Thrombogenicity Index (IT)

$$IT = \frac{C14:0 + C16:0 + C18:0}{(0.5\ x\ \sum MUFA) + (0.5\ x\ \sum n6) + (3\ x\ \sum n3) + \frac{\sum n3}{\sum n6}}$$

IT shows the tendency to form clots in the blood vessels. It relies on the data about the effect of

several fatty acids on the plasma thrombogenic potential, indicating the tendency to form clots in the blood vessels. It represents the relationship of pro-thrombogenic FAs (C12:0, C14:0, and C16:0) and anti-thrombogenic FAs (MUFAs and the PUFA n-3 and PUFA n-6 families. Hence, the consumption of foods with low IT is beneficial for CVH Cardiovascular Health (CVH) [24].

## 3. Data analysis

Data were analyzed using MS-Excel and the Statistical and Data Analysis Software Package: Minitab 17\18 (State College, PA: Minitab, Inc). Collected and obtained data were expressed as means ± SD for experiments independently. Shapiro Wilk test was performed on the data set to assess the normality of the distribution. A one-way analysis of variance (ANOVA) was performed to compare the difference in the respective mean values among the groups for all the assays followed by multiple comparison tests. Regression analysis was applied for the LWA (Length-weight Analysis). A P-value of $< 0.05$ was considered statistically significant.

## 4. Results

### 4.1. Characterization of synthesized Se nanoparticles

Physico-chemical analysis revealed that the Se-Nps showed an average size distribution of 30–50 nm, and 40 ± 12 nm in aqueous solution (Fig 1).

The average particle size determined by Scherrer's equation was about 31 nm. The diffraction peak of Se-Nps obtained was well defined and was in agreement with the standard XRD patterns of Se-Nps (JCPDS-06-0362). Zeta potential analysis revealed the Se-Nps had a potential of -50 ± 07 mV indicating high stability in aqueous solution and aggregation effect in aqueous solution (Milli-Q water) [22].

### 4.2. Survivability and well being

The mean survivability of tilapia was 100%, 97.62%, 95.37%, and 54.73% for the control group and dietary Se-Nps supplemental doses of 0.5, 1.0 and 2.0 mg/kg, respectively (Table 2).

The survival rate was the lowest in the 2 mg/kg inclusion, compared to other groups ($p < 0.05$). The well-being of the fish was determined as Fulton's condition factor for each treatment. The condition factor, K for the *O. niloticus* is shown in Table 3. The condition factor, K computed were 2.40, 2.40, 2.68, and 2.39 respectively for fish fed on 0, 0.5, 1.0, and 2.0

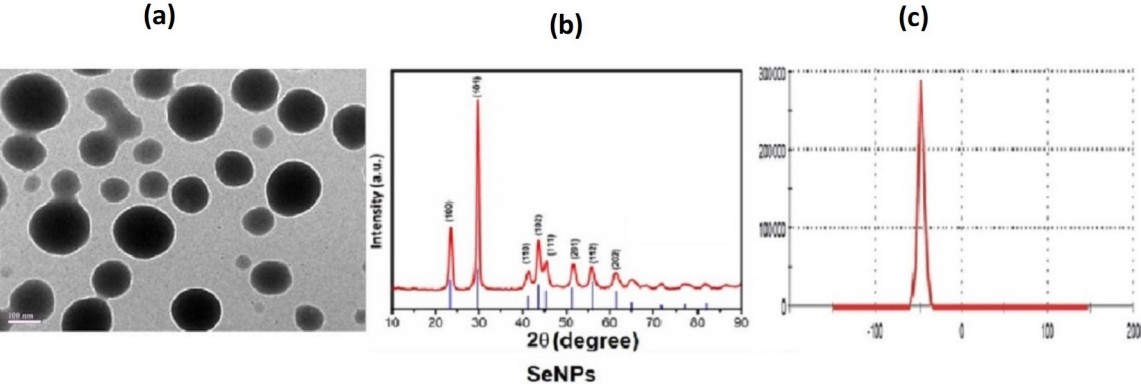

**Fig 1.** Physicochemical characterization showing the representative images of TEM **(a),** XRD **(b)** and zeta potential **(c)** for assessing the stability in solution for Se-Nps [22].

**Table 2. Mean values ± standard deviations (SD) and ranges for biometric data of Nile tilapia fingerlings fed on different levels of Se-Nps inclusions.**

| Dietary supplements | Length (cm) | | Weight (gm) | | Percentage Survivability |
|---|---|---|---|---|---|
| | Mean ± SD | Range | Mean ± SD | Range | |
| Control | 6.81 ± 0.69 | 5.70–7.90 | 7.57 ± 1.68 | 4.60–10.10 | 100.00 |
| 0.5 mg/Kg | 7.03 ± 0.13 | 5.50–8.00 | 8.36 ± 1.66 | 4.08–11.08 | 97.62 |
| 1 mg/Kg | 8.42 ± 0.40 | 7.30–9.10 | 16.09 ± 2.28 | 12.00–19.79 | 95.37 |
| 2 mg/Kg | 5.82 ± 0.43 | 5.11–6.90 | 4.74 ± 1.04 | 3.08–7.00 | 54.73 |

mg/Kg Se-Nps. The highest K value was observed for the 1 mg/kg treatment and the lowest was that obtained for the 2 mg/Kg treatment (Table 3).

## 4.3. Length-weight relationship dynamics & growth performance

The mean lengths and weights were significantly high in the 1 mg/kg inclusion group compared to other groups (p < 0.05, Table 2). On the contrary, significantly weak growth outcomes were obtained in the 2 mg/kg inclusion group. Statistical analysis revealed significant variations (p < 0.05) in the average weight and length among all the treatments (Table 3). The value of the regression coefficient obtained from the LWR was 0.74, 0.89, 0.91, and 0.68 for fish fed on different dietary Se-Nps inclusion levels 0.0 (control), 0.5, 1.0, and 2.0 mg/kg, respectively. There was a significant correlation between length and weight (p < 0.05). All the treatment groups showed a steady increase in weight following exposure to their respective treatments. Significantly, increased mean weights were obtained in fish fed with 1 mg Se-Nps, followed by 0.5 gm Se-Nps in comparison to the other groups (p < 0.05). As presented in Table 2, the results of the current study showed that all the treatments, except 1 mg/kg, and the control groups demonstrated negative allometric growth (b < 2.5). Fish exposed to 1 mg/kg Se-Nps inclusions gave a "b" value close to 3 (b > 2.5) representing isometric growth. The growth performance of *O. niloticus* differed significantly (P < 0.05) in terms of weight, RGR, SGR, FI and PER (Table 3). Feed conversion ratios (FCR) were assessed based on the feed consumption per treatment (considering the live fish in each treatment) throughout the whole research period. Treatment at 1 mg Se-Nps/kg showed higher RGR (1576.04), and SGR (4.70). Relatively poor FCR was observed in the control group compared to the other treatments; the highest FCR (1.91) being recorded for the 1 mg Se-Nps/kg inclusion (Table 3).

## 4.4. Proximate composition of fish body

The average protein content of the basal feed used for all the treatments was 50.72%. Proximate composition analysis of the control and treated fish groups is presented in Table 4. Body moisture, crude protein, crude lipid, crude fiber, and ash did not differ significantly among the treatments. However, the protein content for all the treatments was high, and that for fish

**Table 3. Coefficient of determination ($R^2$), regression of coefficient (b), condition factor (K), relative growth rate (RGR in %), specific growth rate (SGR in %), feed conversion rate (FCR), feed intake (FI) and protein efficiency ratio (PER) for the experimented Nile tilapia fingerlings fed on different levels of Se-Nps inclusions.**

| Dietary Supplements | $R^2$ | b | RGR (%) | SGR (%) | FCR | FI | PER | K |
|---|---|---|---|---|---|---|---|---|
| Control | 0.74 | 2.08 | 680.41 ± 0.15 | 3.42 ± 0.19 | 1.31 ± 0.04 | 4.38 ± 0.31 | 1.89 ± 0.05 | 2.35 |
| 0.5 mg/Kg | 0.89 | 2.35 | 770.83 ± 0.15 | 3.61 ± 0.20 | 1.18 ± 0.03 | 4.68 ± 0.30 | 1.87 ± 0.05 | 2.37 |
| 1 mg/Kg | 0.95 | 2.81 | 1576.04 ± 0.18 | 4.70 ± 0.23 | 1.91 ± 0.03 | 7.16 ± 0.42 | 2.80 ± 0.05 | 2.78 |
| 2 mg/Kg | 0.68 | 2.46 | 383.67 ± 0.20 | 2.63 ± 0.21 | 1.22 ± 0.07 | 3.23 ± 0.38 | 1.84 ± 0.06 | 2.26 |

**Table 4. Proximate composition (mean ± SD) of fish whole body at the initial and end of the experiment for the different treatment types.**

| Variables | Initial experiment | Control | 0.5 mg/kg | 1 mg/kg | 2 mg/kg |
|---|---|---|---|---|---|
| MOISTURE | 79.69 ± 2.32 | 73.23 ± 2.02 | 73.01 ± 2.01 | 75.04 ± 2.03 | 74.15 ± 2.32 |
| CRUDE FIBER | ND | ND | 0.40 ± 0.002 | 0.17 ± 0.001 | 0.10 ± 0.0006 |
| CRUDE FAT | 0.12 ± 0.002 | 3.92 ± 0.32 | 3.64 ± 0.33 | 3.74 ± 0.31 | 3.07 ± 0.32 |
| CRUDE PROTEIN | 10.47 ± 0.91 | 13.71 ± 1.2 | 13.37 ± 0.96 | 14.2 ± 1.01 | 13.35 ± 1.1 |
| ASH | 3.68 ± 0.003 | 2.69 ± 0.002 | 2.66 ± 0.002 | 2.20 ± 0.001 | 2.85 ± 0.002 |
| CHO | 5.05 ± 0.32 | 4.45 ± 0.3 | 5.32 ± 0.24 | 5.22 ± 0.21 | 5.58 ± 0.3 |

exposed to 1 mg Se-Nps showed relatively higher protein content than that of other treatments.

### 4.5. Level of selenium

The level of selenium in the fish blood, muscle, and liver following the results were obtained using Flameless Atomic Absorption Spectrophotometry (FAAS). Significant differences were observed in the levels of Se in the different inclusions of blood and liver tissue among 2 mg/kg supplemented group compared to other groups ($p < 0.05$) (Table 5), which followed an increasing trend through increment with an increase in Se concentrations.

### 4.6. Fatty acid analysis

**4.6.1. Fatty acid profile analysis.** GC-MS analysis revealed the distribution of 22 fatty acids found in the tilapia samples, of which seven were SAFAs, five were MUFAs, and five were PUFAs (Table 6). The results were expressed in terms of the relative peak area/percentage of abundance. PUFA content was higher than the MUFA and the SAFA for all the experimental and the control groups. Importantly, omega -3 fatty acid was significantly high ($\Sigma$ n-3 = 9.05) in 1 mg/kg Se-Nps inclusion group compared to that of any other treatments including the control group (Fig 2), whereas omega-6 fatty acids were lower ($\Sigma$ n-6 = 27.42) in 1 mg/kg Se-Nps inclusion group in comparison to other groups (Fig 2, Table 6).

The total PUFA content was significantly high (38.47, $p < 0.05$) in 1 mg/kg Se-Nps inclusion group, while SAFA content (26.03) was significantly low ($p < 0.05$) in 1 mg/kg Se-Nps group compared to other groups (Table 6).

These findings were further confirmed by the mass spectrogram, including DHA, 22:6 (n-3) and (EPA), 20:5 (n-3) as the representative omega-3 fatty acids, and AdA 22:4 (n-6) and DGLA 20:3 (n-6) as the representative omega-6 fatty acids, which depicted the abundance of the n-3 and n-6 fatty acids was considerably high and low in the 1 mg Se-Nps group, respectively (Figs 3–6).

**4.6.2. Nutritional quality indexes (NQI).** Different fats and fatty acids have varying quality indices for assessing their dietary benefits. In this study, a combination of indexes was utilized to conduct a comprehensive evaluation of the nutritional values of the fish tissue after

**Table 5. Level of selenium (mean ± SD) in the liver, muscle tissue, and blood of fish at the end of the experiment for the different types of Se-Nps inclusions and the control group.**

| | Control | 0.5 mg/Kg | 1 mg/Kg | 2 mg/Kg |
|---|---|---|---|---|
| LIVER | 0.61 ± 0.001 | 0.7 ± 0.001 | 0.77 ± 0.001 | 0.86± 0.002 |
| TISSUE | 0.07 ± 0.0002 | 0.11 ± 0.002 | 0.11 ± 0.0001 | 0.11 ± 0.001 |
| BLOOD | 0.011 ± 0.0001 | 0.016 ± 0.001 | 0.016± 0.0002 | 0.04± 0.0002 |

**Table 6. Percentage of abundance of the FAs with reference to the contribution of the essential fatty acid in accordance with the saturation groups within and across the treatment groups.**

| Formula | Saturation (C:D form) | Fatty Acid Groups | % Peak Area\Abundance | | | |
|---|---|---|---|---|---|---|
| | | | Control | 0.5 mg | 1 mg | 2 mg |
| C13H26O2 | C12:0 | SAFA | 0.23 | 0.15 | 0.1 | 0.29 |
| C15H30O2 | C14:0 | SAFA | 1.99 | 1.8 | 1.7 | 1.98 |
| C16H32O2 | C 15: I | SAFA | 1.32 | 0.44 | 0.39 | 1.42 |
| C16H32O2 | C15:0 | SAFA | 0.69 | 0.74 | 0.4 | 0.42 |
| C17H34O2 | C 16:0 | SAFA | 19.97 | 18.37 | 18.21 | 19.99 |
| C17H32O2 | C 16: 1(n7) | MUFA | 6.34 | 4.85 | 4.73 | 6.6 |
| C18H36O2 | C17:0 | SAFA | 0.81 | 0.9 | 0.78 | 0.87 |
| C19H38O2 | C18:0 | SAFA | 6.74 | 6.61 | 5.45 | 6.99 |
| C19H36O2 | C 18:1 *cis(n9)* | MUFA | 20.31 | 22.46 | 23.87 | 23.38 |
| C19H36O2 | C 18:1 *trans(n9)* | MUFA | 3.75 | 3.45 | 3.19 | 3.22 |
| C19H34O2 | C 18:2 *(n6)* | PUFA | 27.57 | 25.17 | 24.71 | 26.49 |
| C19H32O2 | C 18: 3 (n3) | PUFA | 3.44 | 3.55 | 3.65 | 2.33 |
| C21H40O2 | C 20: 1 (n9) | MUFA | 0.55 | 0.68 | 0.86 | 0.84 |
| C21H36O2 | C 20: 3 (n6) | PUFA | 1.13 | 1.11 | 1.09 | 1.27 |
| C21H34O2 | C 20: 4 (n6) | PUFA | 1.71 | 1.63 | 1.06 | 1.99 |
| C22H36O2 | C20: 4 (n3) | PUFA | 0.25 | 0 | 0.33 | 0.33 |
| C21H32O2 | C 20: 5 (n3) | PUFA | 0.28 | 0.29 | 0.29 | 0.21 |
| C23H38O2 | C 22:4 (n6) | PUFA | 0.52 | 0.52 | 0.26 | 0.59 |
| C23H36O2 | C 22: 5 (n6) | PUFA | 0.65 | 0.64 | 0.3 | 0.88 |
| C23H36O2 | C 22: 5 (n3) | PUFA | 0.2 | 1.16 | 1.21 | 1.01 |
| C23H34O2 | C 22: 6 (n3) | PUFA | 2.98 | 3.35 | 3.57 | 0.97 |
| **Unidentified** | | | 1.03 | 1.16 | 0.94 | 1.02 |
| **Σ SAFA** | | | 31.75 | 29.01 | 26.03 | 31.96 |
| **Σ MUFA** | | | 31.95 | 31.44 | 32.65 | 34.04 |
| **Σ PUFA** | | | 34.73 | 35.42 | 38.47 | 32.07 |
| **Σ n-6** | | | 31.58 | 29.07 | 26.42 | 31.22 |
| **Σ n-3** | | | 7.15 | 7.35 | 9.85 | 4.85 |

being exposed to different Se-Nps inclusion levels (Fig 7). Ratios of PUFA/SAFA and n-3/n-6 were found to be 1.35 and 0.33, respectively in the fish fed 1 mg/kg Se-Nps inclusion, which was relatively higher compared to other treatments, including the control group. In addition, fish fed 1 mg/kg Se-Nps exhibited lower ratios of n-6/n-3 (3.03). The atherogenicity index IA, and thrombogenicity index IT were relatively low at 0.36 and 0.44, respectively in the 1 mg/kg Se-Nps inclusion group compared to other treatments including the control (Fig 7).

## 5. Discussion

Aquaculture enhancement is currently been pursued in different dimensions for addressing the needs of malnutrition and providing adequate nutrition to the entire community, which involves the inclusion of nano-nutrients, fermented materials, trace elements, and many others [25]. The use of nanoparticles of trace elements is one of the promising approaches for feed supplementation in a variety of fish, which serves as a handy way of delivery and easy availability to the fish. The present study showed that the administration of a specific dose of Se-Nps enhanced the growth performance and nutrient quality index in Nile tilapia that serves as one of the staple fish food across the continents and can meet the nutrition requirement of the

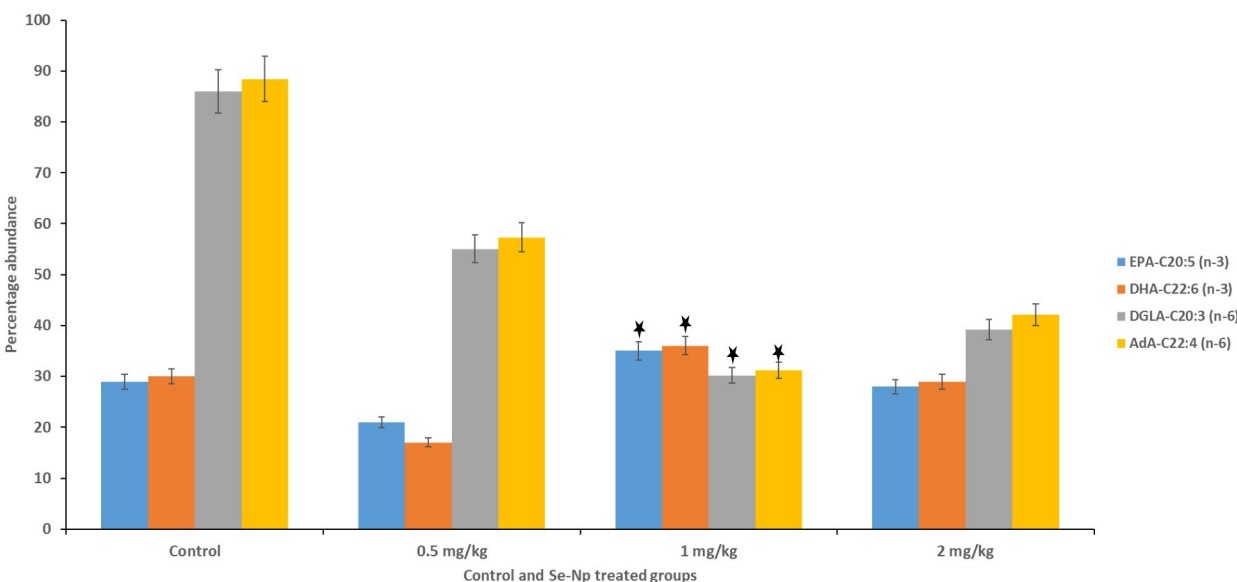

**Fig 2. Percent abundance of representative essential omega 3 (n-3; DHA, docosahexaenoic acid and EPA, eicosapentaenoic acid) and omega 6 (n-6; DGLA, eicosatetraenoic acid and AdA, eicosatrienoic acid) fatty acids among all the groups, control; 0.5 mg Se-Nps; 1 mg Se-Nps; 2 mg Se-Nps groups, depicting 1 mg/kg Se-Nps group possessing significantly high n-3 fatty acids and low n-6 fatty acids (p < 0.05, ANOVA test), in comparison to other treatments.**

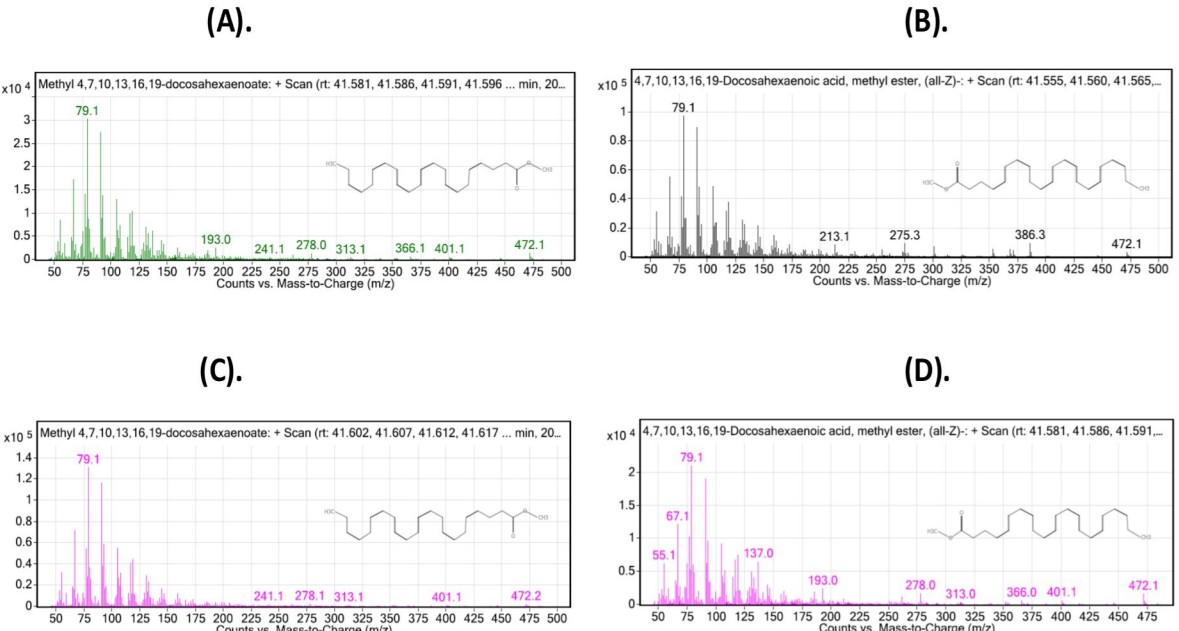

**Fig 3.** Mass Spectrometry Plot: Counts vs. mass-to-charge (m/z) assessed using GC/MS Mass hunter: Representative plot for omega-3 depicts DHA, C 22:6 (n-3), for the different treatments: (A), Control; (B), 0.5 mg/kg Se-Nps; (C), 1 mg/kg Se-Nps; (D), 2 mg/kg Se-Nps inclusions. The different treatment samples for the given fatty acid depicted most of the characteristic indexed peaks for the fatty acids, with highest peaks as the main representative of the fatty acid with similar molecular ion peak (m/z).

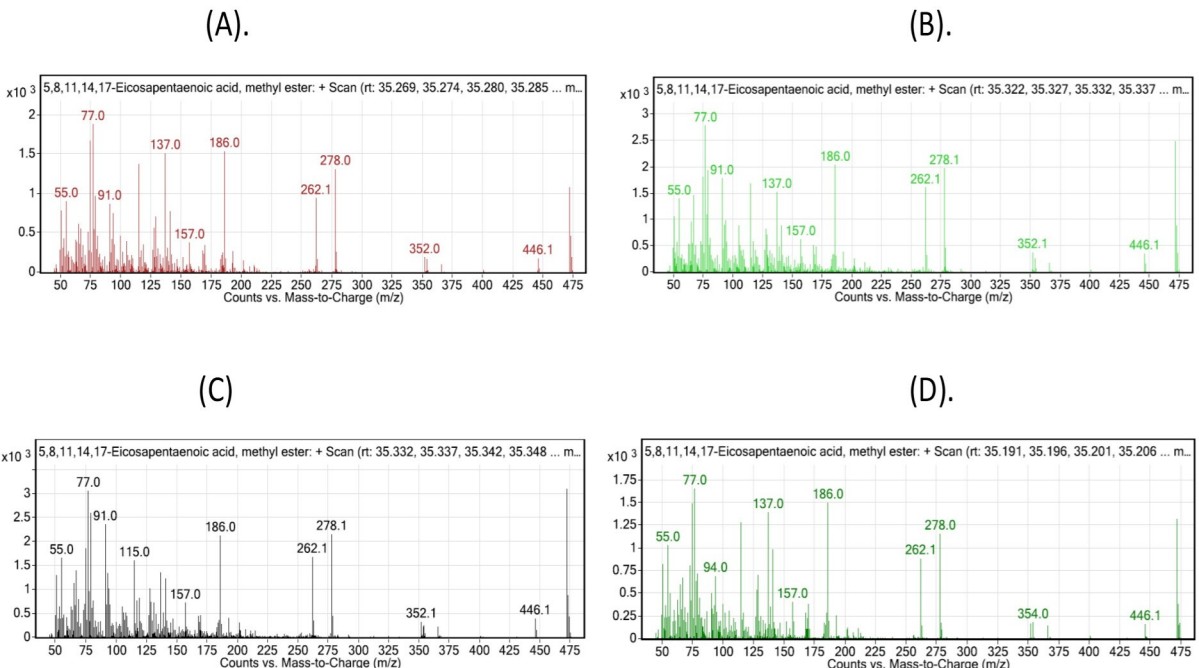

**Fig 4. Mass Spectrometry Plot: Counts vs. mass-to-charge (m/z) assessed using GC/MS Mass hunter:** Representative plot for omega-3 depicts EPA, C 20:5 (n-3) for the different treatments: (A), Control; (B), 0.5 mg/kg Se-Nps; (C), 1 mg/kg Se-Nps; (D), 2 mg/kg Se-Nps inclusions. The different treatment samples for the given fatty acid depicted most of the characteristic indexed peaks for the fatty acids, with the highest peaks as the main representative of the fatty acid with a similar molecular ion peak (m/z).

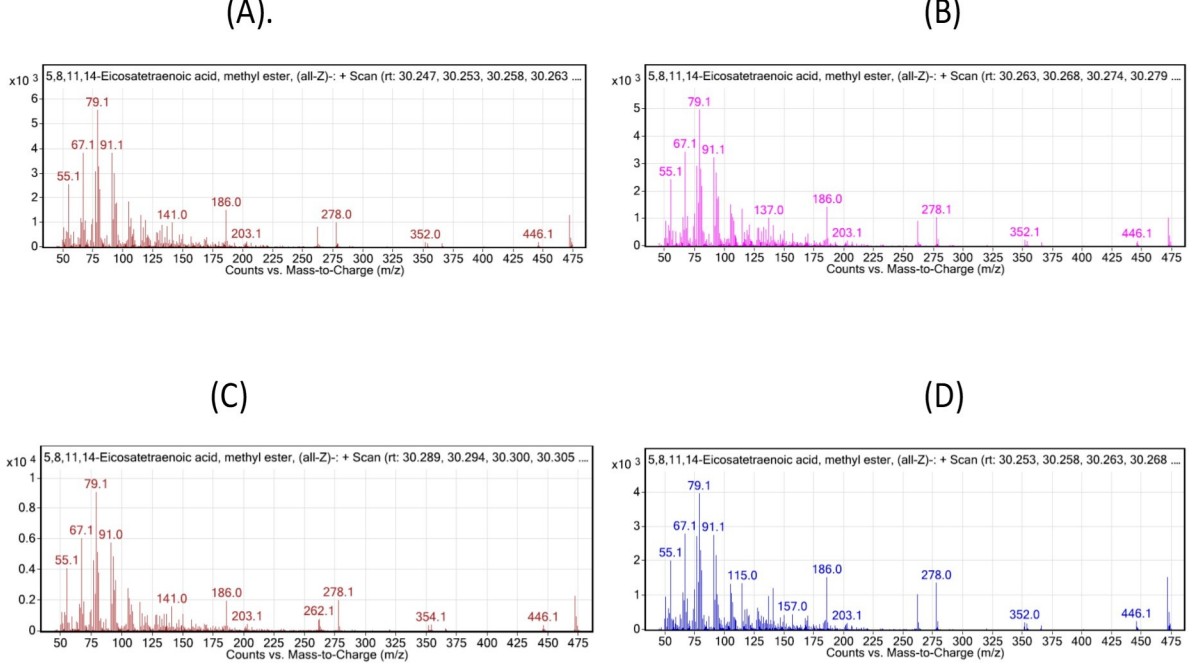

**Fig 5. Mass Spectrometry Plot: Counts vs. mass-to-charge (m/z) assessed using GC/MS Mass hunter:** Representative plot for omega-3 I depicts AdA, 20:4 (n-6) for the different treatments: (A), Control; (B), 0.5 mg/kg Se-Nps; (C), 1 mg/kg Se-Nps; (D), 2 mg/kg Se-Nps inclusions. The different treatment samples for the given fatty acid depicted most of the characteristic indexed peaks for the fatty acids, with highest peaks as the main representative of the fatty acid with similar molecular ion peak (m/z).

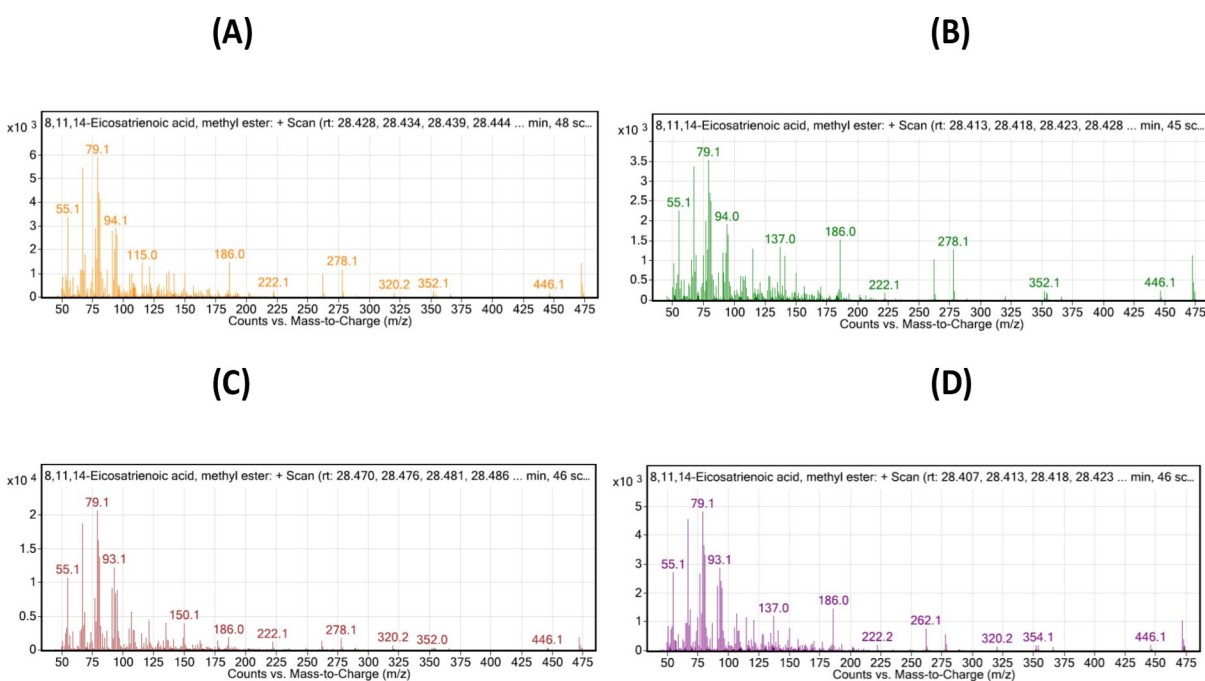

**Fig 6.** Mass Spectrometry Plot: Counts vs. mass-to-charge (m/z) assessed using GC/MS Mass hunter: Representative plot for omega-3 depicts DGLA, 20:3 (n-6) for the different treatments: (A), Control; (B), 0.5 mg/kg Se-Nps; (C), 1 mg/kg Se-Nps; (D), 2 mg/kg Se-Nps inclusions. The different treatment samples for the given fatty acid depicted most of the characteristic indexed peaks for the fatty acids, with the highest peaks as the main representative of the fatty acid with a similar molecular ion peak (m/z).

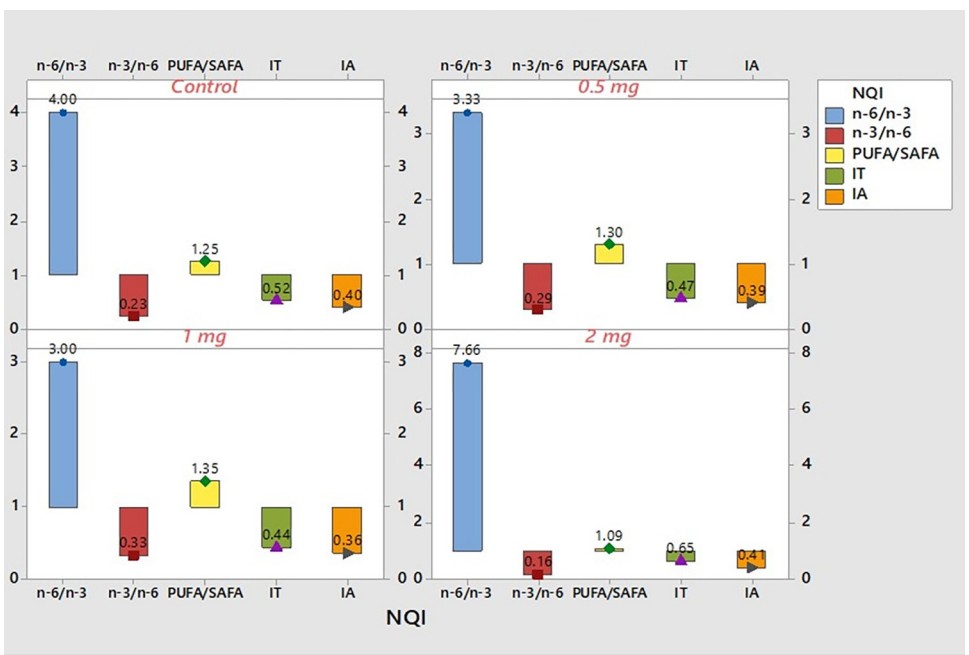

**Fig 7. Nutritional quality indexes (NQI): n-6/n-3, PUFA/SAFA.** IT, and IA for Control (A), 0.5 mg Se-Nps/kg (B), 1 mg Se-Nps/kg (C), and 2 mg Se-Nps/kg (D) inclusions.

growing population. In the present study, the experimental and the control groups showed negative allometric growth (b < 3); however, tilapia fish fed 1 mg/kg of Se-Nps inclusion showed nearly isometric growth with b value of 2.81. Previous studies have reported significantly higher growth in terms of weight gain in the diet containing 1 mg/kg of Se-Nps in fish, which was in line with our findings [25].

The present study revealed that 1 mg/kg Se-Nps supplementation resulted in significantly higher weight gain (> 210%, p > 0.05) compared to other groups, which was in line with other studies performed in crucian carp [26] and rainbow trout [27]. Selenium is a vital and key element of fish nutrition and is essential for the functioning of an array of Se-carrying proteins, called sialoproteins [28, 29]. Nanoselenium has improved bioavailability because of its extraordinarily small size, such that it can be easily absorbed and be readily uptaken by the cells, thereby enhancing the metabolic and physiological activities [30].

Nile tilapia demonstrated enhanced growth performance upon nanoselenium supplementation. Growth parameters RGR and SGR values were significantly high and low in the 1 mg/kg and 2 mg/kg Se-Nps inclusion groups, respectively in comparison to others. The low growth performance in 2 mg/kg Se-Nps inclusion group could be attributed to the toxic effect of the over-dose supplementation of Se-Nps. In addition, the fish fed 1 mg/kg Se-Nps revealed better k values (condition factor) than the other treatment groups. Several studies also reported low FCR, SGR, and RGR obtained for 2 mg/kg Se-Nps in different fish varieties [5, 31]. Fish fed 1 mg/kg Se-Nps had by far lower FCR (1.07) than the control group (2.53) ($p < 0.05$). According to [5], the higher the FCR, the lower is the growth efficiency and inverse is the link between SGR and FCR. The result from our study is in agreement with these findings [5].

However, some studies reported enhanced growth performance at higher Se-Nps supplementation dose, such as at 3.67 mg/kg Se inclusion in African catfish [32]. On the contrary, optimum Se-Nps dose concentration was low for rainbow trout (0.38 mg/kg) [33], channel catfish (0.25 mg/kg) [34], and for groupers (0.77 mg/kg) [35]. Such findings demonstrate that the optimum dose of Se-Nps supplementation for enhancing growth and productivity varies with fish type [36], which could be attributed to the bioaccumulation or bioavailability of Se inside the fish tissues.

Selenium distribution varied across different tissues in the fish treatments. Se accumulation in the fish tissues was generally low and increased in a concentration-dependent manner. Se levels in the body tissues of Nile tilapia was 0.055 mg/kg in our study, which was lower than that obtained for catfish (0.87 mg/kg) [36]. This finding was in agreement with other studies that showed selenium is not toxic upto 1.12 mg/kg in tilapia [37]. In addition, a slightly higher and dose-dependent increment of Se levels was observed in the liver with an increase in inclusion concentration. Se level in the blood samples were similar for the 0.5 and the 1 mg/kg inclusions, while Se level increased significantly in the 2 mg/kg treatment group. The observed difference in the accumulation of Se within the particular organ/tissue could be attributed to the difference in their physiological functions and increased dose of Se-Nps exposure. The liver is the most important target organ involved in the metabolism and detoxification of xenobiotics in vertebrates [38]. Because fish were not starved prior to sampling, livers were observed to contain high fat [39]. The low Se level in the blood samples could be because blood is involved in transporting, and not accumulating Se, from the digestive system to other organs, specifically to the liver, wherein accumulation and bio-magnification of selenium [33, 40] occurs.

Another intriguing feature of growth performance was the content of different fatty acids (FA) in the fish. Among the different FAs analyzed, the SAFA C 16:0 and C 18:0, MUFA C 18: 2 (n6), C 18:1 (cis n9), C 18: 1 (trans n9) and C: 16: 1 (n7), PUFA C 18:2 (n6), C 18: 3 (n3), and C 22: 6 (n3) showed varying percentages of abundance across the treatment groups (Fig 4).

Previous studies suggest that diets rich in saturated fatty acids are not healthy, and the consumption of mono and polyunsaturated fatty acids is recommended to maintain good health [41, 42]. Generally, the good thing is fish species have a relatively low SAFA [43], which have been linked to coronary heart disease [44]. Many studies have shown that fish oil has a high nutritional value because it is rich in PUFAs, which consist of n-3 PUFAs and n-6 PUFAs, particularly docosahexaenoic acid (DHA) and eicosapentaenoic (EPA) [45–47]. In this study, the fatty acids were observed in order: PUFA > MUFA> SAFA, in mainly the 1 mg/kg Se-Nps inclusion group, which could be benefitting for the health of humans. PUFA/SAFA and n3/n6 ratios are considered to be important parameters correlated to the nutritional value of food types for human health. According to the WHO report, the minimum value for PUFA: SAFA is 0.45 or more [48, 49]. Our findings revealed overall PUFA:SAFA greater than 0.45, which was further significantly high in the fish fed 1 mg/kg Se-Nps inclusion compared to other groups.

In the present study, fish fed with 1 mg/kg Se-Nps inclusion showed higher levels of DHA and EPA, in addition to significantly high omega-3 content (n-3 = 9.37) in comparison to other treatments and control, which could be thus attributed to the effect of Se-Nps supplementation at the optimum concentration (1 mg/kg). Such findings were in agreement with previous studies, which reported that Se-Nps was effective in improving the productive performance and glutathione peroxide (GSH-Px) activity of enzymes responsible for improving the fatty acid profile and antioxidant potential [50–53]. In addition, n-3/n-6 ratios, which depict the nutritional quality indices, were relatively low in the treatment groups of this study, which were in line with previous studies [54], and have been shown to possess health-promoting effects in Nile Tilapia. Studies have shown that Nile tilapia exhibits growth depression when fed with diet rich in n-3 PUFAs and low in n-6 PUFA. Dietary n-3 PUFAs such as linolenic acid at 0.4–0.6%, along with higher amount of n-6 PUFAs such as linoleic acid enhanced the growth of juvenile tilapia; however, higher levels of linolenic acid had adverse effects of the growth of Nile tilapia, thereby indicating that a balanced ratio of n-3/n-6 PUFAs (i.e., amount of n-3 PUFA < amount of n-6 PUFA) is essential for the overall growth and productivity [55, 56]. NQI: IA and IT were considerably low in 1 mg/kg Se-Nps inclusion group, which corroborated with previous studies [57, 58]. Generally, fish fed with 1 mg/kg Se-Nps inclusion demonstrated highest weight and length gain, exhibited high nutritional quality meat, in addition to higher levels of n-3 fatty acids, all of which could be because of the optimum dose of Se-Nps supplementation leading to higher bioavailability and metabolism in tilapia (*O. niloticus*).

Overall, the study revealed the growth-promoting effects and the improved nutritional quality of Nile tilapia upon nanoselenium supplementation. Nanonutrient supplementation is considered as one of the most promising approaches for enhancing aquaculture in recent time, and this study has elaborated such effects using Se-Nps in Nile tilapia, wherein it was found that Se-Nps treated Nile tilapia at specific dose exhibited higher levels of PUFAs that will aid in the overall productivity of fish. Such application of nanonutrients could be expanded to a variety of commercial fish for improving the overall well-being of the fish, which could further address the issues of malnutrition and protein requirement across the global population.

## 6. Conclusion

Nanoselenium supplementation enhanced the growth performance and nutrition quality of Nile tilapia at a specific doses. Selenium exhibits diverse roles in the productivity and immune status of fish; hence, nanoselenium supplementation increased the bioavailability and overall growth parameters like length, weight, omega 3 fatty acid content, and overall growth of

tilapia. The findings highlight the utility of nanotechnology for the enhanced production of Nile tilapia that could be utilized for meeting the increasing demand of malnutrition across the global population. Industries related to fish farming could explore the development of nano-feed supplementations with nanoselenium and other micronutrients at optimum levels for enhancing the growth performance and productivity.

## Acknowledgments

The authors extend their gratitude to the laboratory and technical support of the Ethiopian Biotechnology Institute (EBTI) and to the deanship of scientific research of Taif University Researchers Supporting Project number (TURSP-2020/203), Taif University, Taif, Saudi Arabia for accomplishment of the research work.

## Author Contributions

**Conceptualization:** Fasil Dawit Moges, Abebe Getahun Gubale, Biswadeep Das.

**Data curation:** Fasil Dawit Moges, Hamida Hamdi, Abebe Getahun Gubale, Biswadeep Das.

**Formal analysis:** Fasil Dawit Moges, Hamida Hamdi, Abeer Abu Zaid, Abebe Getahun Gubale.

**Investigation:** Fasil Dawit Moges, Hamida Hamdi, Amal Al-Barty, Abeer Abu Zaid, Abebe Getahun Gubale.

**Methodology:** Fasil Dawit Moges, Hamida Hamdi, Amal Al-Barty, Manisha Sundaray, S. K. S. Parashar, Abebe Getahun Gubale.

**Project administration:** Hamida Hamdi, Amal Al-Barty, Abeer Abu Zaid, Abebe Getahun Gubale, Biswadeep Das.

**Resources:** Abebe Getahun Gubale, Biswadeep Das.

**Supervision:** Hamida Hamdi, Abebe Getahun Gubale, Biswadeep Das.

**Validation:** Abeer Abu Zaid, Manisha Sundaray, Biswadeep Das.

**Writing – original draft:** Fasil Dawit Moges, Biswadeep Das.

**Writing – review & editing:** Fasil Dawit Moges, Amal Al-Barty, Abeer Abu Zaid, Manisha Sundaray, Biswadeep Das.

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
