## [Decision Letter · Decision Letter 0]

16 Mar 2022

PONE-D-22-01541Nano-Selenium supplementation improves the growth performance in Nile tilapia (Oreochromis niloticus) through the enhancement of w-3 fatty acids and nutrition quality indexPLOS ONE

Dear Dr. Das,

Thank you for submitting your manuscript to PLOS ONE. After careful consideration, we feel that it has merit but does not fully meet PLOS ONE’s publication criteria as it currently stands. Therefore, we invite you to submit a revised version of the manuscript that addresses the points raised during the review process.

We look forward to receiving your revised manuscript.

Kind regards,

Mahmoud A.O. Dawood, PhD

Academic Editor

PLOS ONE

Journal Requirements:

[The authors extend their appreciation to the deanship of scientific research for funding this article by Taif University Researchers Supporting Project number (TURSP-2020/203), Taif University, Taif, Saudi Arabia. In addition, the support of the Ethiopian Biotechnology Institute (EBTI) is gratefully acknowledged for assistance for the research work.]

 [The author(s) received no specific funding for this work.]

Reviewers' comments:

Reviewer's Responses to Questions

**Comments to the Author**

1. Is the manuscript technically sound, and do the data support the conclusions?

Reviewer #1: Yes

Reviewer #2: Partly

2. Has the statistical analysis been performed appropriately and rigorously? 

Reviewer #1: Yes

Reviewer #2: No

3. Have the authors made all data underlying the findings in their manuscript fully available?

Reviewer #1: No

Reviewer #2: Yes

4. Is the manuscript presented in an intelligible fashion and written in standard English?

Reviewer #1: Yes

Reviewer #2: Yes

5. Review Comments to the Author

Reviewer #1: The findings of the present study "Nano-Selenium supplementation improves the growth performance in Nile tilapia (Oreochromis niloticus) through the enhancement of w-3 fatty acids and nutrition quality index " are new and practical. I would recommend it for publication after some revisions. Please check the attached file where I tracked the changes and added a comment.

Reviewer #2: Manuscript evaluation (PONE-D-22-01541)

Overall assessment

The objective of the study was “Nano-selenium supplementation improves the growth performance in Nile tilapia (Oreochromis niloticus) through the enhancement of w-3 fatty acids and nutritional quality index”. Indeed a good objective chosen for the improvement of aquaculture of this fish. Overall, the manuscript provides a significant amount of knowledge to the existing literature. However, there are some serious concerns regarding construction of the manuscript. I found the presentation of the manuscript is little confusing and the narrative of the manuscript was missing due to the repetition of "fact" in many sections. Although very interesting to read, it will be improved if performed major changes throughout the manuscript.

Title: Needs revision it should be modified as “Effects of particle selenium on growth performance and nutritional quality in Nile Tilapia, Oreochromis niloticus.

Abstract:

Abstract needs revision as some critical information is still missing like FCR, SGR, PER etc.

L45-47: not clear needs revision

L52: what was the weight and size of the fingerlings?

L58: condition factor “C” or “K”?

L63: Np or Nps?

Keywords: why nanotechnology? Keywords do not represent title/theme of the manuscript.

Introduction:

L 87: aquatic fish?

L98-102: not related to this study and needs revision

L102-11: again majority of this text not related to aquaculture. I suggest incorporation of a paragraph emphasized on aquaculture nutrition not feeding or histology of the species as it is well known fact.

P4 L 119: delete ‘an’ before ‘SFA’

P4 L124-128: again the paragraph is related to human not fish. Rewrite it.

Materials and Methods:

P5 L167-168: how the water quality was maintained?

P6 L181: why the hatchling was called as fingerlings? What was the size of the fingerlings? How many days old fingerlings were used in the study? Add these informations.

L183: Fries/fingerlings? How length weight relationship was calculated in such a small fish I doubt.

L189: add protein and energy content of each diet.

L197-199: feed was fed twice a day? See next line ‘after every hour of feeding”?

P6 L202-205: Estimation of LWR in such a small fish species not possible. Need justification/or support in the form of citations.

P7 L236: how the blood was collected from the gills?

P238: add ‘and’ after ‘fish’.

P8 L 265-268: why only these parameters were selected to determine nutritional quality of the fish? Why protein and minerals was not considered?

Table 2: Title not written scientifically, needs revision.

Add feed intake of each group.

Why SD not mentioned in these parameters?

PER and body protein deposition which are considered as the main growth parameters in feeding trials are missing. Please add it.

Why FCR decreased so drastically and increased further? Does selenium affects growth?

P11 L341-346: not in favor of these results as it is redundant in growth study that too such a small fish and short feeding trial as well as small fish.

P12 L371: ‘Use’?

Table 3: proximate composition of initial samples is missing. Add in this table.

Statistical analysis of the data is missing. Also Table is not presented in scientific way.

Check crude fat content at 0.5 mg/kg.

Discussion:

P16 L481-485: Again my concern is about the estimation of LWR in this study. The author should focus on growth data.

P17 L496: Se-Np? Kindly use uniform terms in whole manuscript when representing selenium as nanoparticles like

L499: Se-Np?

L500: N-Se?

506: Se-Nps?

508: Se-Nps?

P17 L514-516: how the present results compared with human Se requirement? I consider comparing results with other fish species.

P18 L545-549: the paragraph may be shifted to introduction section. Not needed here.

P19 L556-562: again these sentences are not required here. The authors have to write significant benefits pertaining to fish.

P19 l570-576: repetition of sentences needs revision.

Conclusion: not sound needs revision.

In my opinion that authors should reframe the manuscript as per suggestions and quarries raised. Presentation needs to be improved throughout the manuscript. The manuscript should be reconsidered after major revision.

6. PLOS authors have the option to publish the peer review history of their article (what does this mean?). If published, this will include your full peer review and any attached files.

Reviewer #1: No

Reviewer #2: **Yes: **Imtiaz Ahmed

---

## [Author Response · Author response to Decision Letter 0]

18 Apr 2022

Reviewer #1: The findings of the present study "Nano-Selenium supplementation improves the growth performance in Nile tilapia (Oreochromis niloticus) through the enhancement of w-3 fatty acids and nutrition quality index " are new and practical. I would recommend it for publication after some revisions. Please check the attached file where I tracked the changes and added a comment. 

We are thankful to the reviewer for reading our manuscript and pointing out scope for improvement. We have thoroughly revised the manuscript and addressed the reviewer comments for improving the overall quality of the paper.

Reviewer #2: Manuscript evaluation (PONE-D-22-01541)

Overall assessment

The objective of the study was “Nano-selenium supplementation improves the growth performance in Nile tilapia (Oreochromis niloticus) through the enhancement of w-3 fatty acids and nutritional quality index”. Indeed a good objective chosen for the improvement of aquaculture of this fish. Overall, the manuscript provides a significant amount of knowledge to the existing literature. However, there are some serious concerns regarding construction of the manuscript. I found the presentation of the manuscript is little confusing and the narrative of the manuscript was missing due to the repetition of "fact" in many sections. Although very interesting to read, it will be improved if performed major changes throughout the manuscript.

We are thankful to the reviewer for reading our manuscript and pointing out scope for improvement. We have thoroughly revised the manuscript and addressed the reviewer comments for improving the overall quality of the paper.

Title: Needs revision it should be modified as “Effects of particle selenium on growth performance and nutritional quality in Nile Tilapia, Oreochromis niloticus.

We have revised the title accordingly as “Effects of selenium nanoparticle on the growth performance and nutritional quality in Nile Tilapia, Oreochromis niloticus.”

Abstract:

Abstract needs revision as some critical information is still missing like FCR, SGR, PER etc.

We have revised the abstract to include all of the above. 

L45-47: not clear needs revision.

We have revised the line accordingly.

L52: what was the weight and size of the fingerlings?

2.5 gm (Av.) in weight and 4 cm (Av.) in length.

L58: condition factor “C” or “K”?

It is “K” not “C”. We have revised throughout the text accordingly.

L63: Np or Nps?

We have revised and used “Nps” throughout the manuscript consistently. 

Keywords: why nanotechnology? Keywords do not represent title/theme of the manuscript.

We have revised the keywords accordingly.

Introduction:

L 87: aquatic fish?

Revised it.

Please omit “aquatic”…it should read “……widely reared fish across…”

We have revised the line accordingly.

L98-102: not related to this study and needs revision

We have revised the line accordingly.

L102-11: again majority of this text not related to aquaculture. I suggest incorporation of a paragraph emphasized on aquaculture nutrition not feeding or histology of the species as it is well known fact.

We have revised the line accordingly.

P4 L 119: delete ‘an’ before ‘SFA’

Revised.

P4 L124-128: again the paragraph is related to human not fish. Rewrite it.

We have rewritten the entire paragraph accordingly.

Materials and Methods:

P5 L167-168: how the water quality was maintained?

Here, we considered the parameters: pH, dissolved oxygen, and conductivity as water quality indicators. 

1st These parameters were checked\\measured within 2 to 3 days per week for the 1st experimental month, until the RAS attain stability, where then after the measurement was done twice per month for the end of the experiment. 

2nd The parameters were adjusted accordingly. As the parameters were dependent each other and mostly work in-cause-effect cyclic manner, adjustment is made accordingly. 

Note: The water quality Parameters were measured using a “Multi-Parameter Water Quality- Meter” which is a sensor with high Precision Water Quality Test Pen Portable Digital (0.01 Accuracy) Instrument. 

P6 L181: why the hatchling was called as fingerlings? 

In our hatchery: 

Healthy Bloodstocks were selected and conditioned for reproduction in 4:1 (Male to Female) ratio. These parents were let to reproduce while fed protein high feed twice per day for 12 days. Eggs were collected carefully from the mouth of female fish. Fertilized and viable eggs were selected and transferred to the hatching set-up. New hatchlings start swimming down to the incubating tanks within 7 days. These yolk sac fries are let fully absorb their yolk sac, where they are transferred in to the rearing tanks and started feeding. These fries are let to grow while fed protein high feed twice per day for 1.5 months. After this stage they were sorted for stocking in to the experimental aquarium (RAS), where they were let to acclimatize for 2 more weeks. New hatchlings (0.02-1 gm) are called “fries” and those between 1 gm and 10 gm are commonly called fingerlings (AFFRIS, 2013)

Reference

AFFRIS (Aquaculture Feed and Fertilizer Resources Information System), FAO, Food and Agriculture Organization, Fisheries and Aquaculture, 2013. Rome, Italy. https://www.fao.org/fishery/affris/species-profiles/nile-tilapia/nutritional-requirements/en/

What was the size of the fingerlings? 

The average size was 2.5 gm in weight and 4 cm in length. 

How many days old fingerlings were used in the study? Add these information.

60 days (2 months)

L183: Fries/fingerlings? How length weight relationship was calculated in such a small fish I doubt.

We have replaced ‘fries’ with ‘fingerlings’ for clarity as Nile tilapia ≥ 2 gm is called fingerling, and the LWR can be measured appropriately in the fingerlings.

L189: add protein and energy content of each diet.

We have added a new table 1 with the above points in accordance with the reviewer suggestions.

Table. Composition, Protein and Energy content of Basal feed. 

No. Ingredient GE CP

 Names % Gram MJ/Kg Kcal/Kg 

1 Fish Meal 35.06 350.60 18.56 4433 60.30

2 Soya Bean 32.54 325.40 17.52 4184 46.47

3 Wheat 18.00 180.00 16.27 3886 13.90

4 Corn 12.00 120.00 17.34 4142 9.48

5 CMC 2.00 20.00 

6 Vitamin Premix 0.40 4.00 

Where, CMC: Carboxymethyl Cellulose, Binder. 

GE: Gross Energy in MJ/Kg (Mega Joules per Kilo Gram), Kcal (Kilo Calorie), CP: Crud Protein. 

L197-199: feed was fed twice a day? See next line ‘after every hour of feeding”?

“Every hour” be substituted by “Every meal” 

P6 L202-205: Estimation of LWR in such a small fish species not possible. Need justification/or support in the form of citations.

The current experiment, studying the effect of the supplementation of dietary nanoparticles, was assessed at the earlier stages of the fish so that to explicitly follow the effect starting from their earlier developmental stages. Observations and data on the toxic effect, survivability, the condition of the fish and other growth parameters like the length-weight relationship were fully covered for the wider range of life stages. The Length-Weight relationship as growth performance parameter in feed, nutrition, and related experiments was assessed in the tilapia fingerlings for obtaining prominent outcomes (fingerlings> 2 gm, length ≥ 4 cm), and which has been reported in various previous studies (Ighwela, et al., 2011, Shahabuddin, et al., 2015, Komba, et al., 2020, Leticia, et al., 2021) utilizing other compounds/agents for assessing growth and productivity.

References 

Félix-Cuencas L, García-Trejo JF, López-Tejeida S, León-Ramírez JJ, Soto-Zarazúa GM. Effect of three productive stages of tilapia (Oreochromis niloticus) under hyper-intensive recirculation aquaculture system on the growth of tomato (Solanum lycopersicum). Latin American Journal of Aquatic Research, 2021; 9(5): 689-701. DOI: 10.3856/vol49-issue5-fulltext-262. 

Ighwela KA, Ahmed AB, Abol-Munafi AB. Condition Factor as an Indicator of Growth and Feeding Intensity of Nile Tilapia Fingerlings (Oreochromis niloticus) Feed on Different Levels of Maltose. American-Eurasian J. Agric. & Environ. Sci. 2011; 11 (4): 559-563. 

Komba EA, Munubi RN, Chenyambuga SW. Comparison of body length-weight relationship and condition factor for Nile tilapia (Oreochromis niloticus) cultured in two different climatic conditions in Tanzania. International Journal of Fisheries and Aquatic Studies. 2020; 8 (3): 44-48. 

Shahabuddin AM, Khan MND, Ayna E, Wonkwon K, Murray WW, Yoshimatsu T, et al. Length-weight Relationship and Condition Factor of Juvenile Nile Tilapia Oreochromis niloticus (Linnaeus 1758) Fed Diets with Pyropia spheroplasts in Closed Recirculating System. Asian Fisheries Science, 2015; 28: 117-129. 

P7 L236: how the blood was collected from the gills?

Blood was collected by using combination of two methods: “Dorsal Aorta Puncture” and Caudal Venous Puncture” procedures so that we fully collect blood samples both from arteries and veins of fish. In both case fish were sedated\\anesthetized using Clove Oil. 

Dorsal Aorta Puncture was done by directly inserting needle on a syringe in to the mouth at the 1st or 2nd gill arch as the fish are small in size. 

P238: add ‘and’ after ‘fish’.

‘and’ is added, it reads ‘ Fish and feed proximate…..’

P8 L 265-268: why only these parameters were selected to determine nutritional quality of the fish? Why protein and minerals was not considered?

Our intent was to evaluate the effect of Selenium on the quality of Lipids. Because many studies have characterized the protein profile and mineral content in Tilapia, so we focussed on characterizing the lipid profile, especially the PUFA content to assess the antioxidant status upon feeding with Se-Nps, because n-3 and n-6 PUFAs have a significant impact on fish growth.

Table 2: Title not written scientifically, needs revision.

We have revised Table 2 and its now new Table 3 with the new data

Table 3. 

Dietary R2 b RGR (%) SGR (%) FCR FI PER K

Supplements 

Control 0.74 2.08 680.41 ± 0.15 3.42 ± 0.19 1.31 ± 0.04 4.38 ± 0.31 1.89 ± 0.05 2.35

0.5 mg/Kg 0.89 2.35 770.83 ± 0.15 3.61 ± 0.20 1.18 ± 0.03 4.68 ± 0.30 1.87 ± 0.05 2.37

1 mg/Kg 0.95 2.81 1576.04 ± 0.18 4.70 ± 0.23 1.91 ± 0.03 7.16 ± 0.42 2.80 ± 0.05 2.78

2 mg/Kg 0.68 2.46 383.67 ± 0.20 2.63 ± 0.21 1.22 ± 0.07 3.23 ± 0.38 1.84 ± 0.06 2.26

Why SD not mentioned in these parameters?

-Mentioned (Table 3 above)

PER and body protein deposition which are considered as the main growth parameters in feeding trials are missing. Please add it.

PER, Protein Efficiency Ratio is presented in table 3.

PER was calculated as:

 PER=FCR*(% FP)/(% PCS)

Where PER: Protein Efficiency Ratio, FCR: Feed Conversion Ratio, FP: feed protein, which is the percentage protein contribution of the treatment feed, PCS: Protein in Culture Species, which is the proximate protein content of Tilapia Fingerlings. 

Why FCR decreased so drastically and increased further? Does selenium affects growth?

Yes, Se affects growth. The effect on growth performance is more significant in its nano form. Se-Nps was found to promote growth up to a certain safe and optimum level (1 mg/kg). 

P11 L341-346: not in favor of these results as it is redundant in growth study that too such a small fish and short feeding trial as well as small fish.

Actually the values: length and weights of the smallest and lengthiest fish are all mentioned on Table 2 (ranges), and are all in favour of the growth performance of the treatments. 

However, for clarity we have omitted the sentences (L341-L344) “Fish fed 1mg/kg Se-Np were observed to comprise the tallest (9.10 cm) and the heaviest (19.79 gm) individual, whereas the smallest fish was 5.11 cm in length and 3.08 gm in weight and found in 2 mg Se-NP supplemented group (Table 1)” in the revised manuscript.

Table 3: proximate composition of initial samples is missing. Add in this table.

We have added this data in the table, which is now new Table 4.

Statistical analysis of the data is missing. 

Added

Also Table is not presented in scientific way. Check crude fat content at 0.5 mg/kg.

We have revised the table 4.

Discussion:

P16 L481-485: Again my concern is about the estimation of LWR in this study. The author should focus on growth data.

The current experiment, studying the effect of the supplementation of dietary nanoparticles, begins the experimentation at the earlier stages of the fish so that to explicitly follow the effect starting their earlier developmental stages. Observations and data on the toxic effect, survivability, the condition of the fish and other growth parameters like the length-weight relationship are fully covered for the wider range of life stages. The Length-Weight relationship as growth performance parameter in feed, nutrition, and related experiments begins with fish samples of younger ages (fingerlings: 1 gm) which is seen in various previous studies (Ighwela, et al., 2011, Shahabuddin, et al., 2015, Komba, et al., 2020, Leticia, et al., 2021,

References 

Félix-Cuencas L, García-Trejo JF, López-Tejeida S, León-Ramírez JJ, Soto-Zarazúa GM. Effect of three productive stages of tilapia (Oreochromis niloticus) under hyper-intensive recirculation aquaculture system on the growth of tomato (Solanum lycopersicum). Latin American Journal of Aquatic Research, 2021; 9(5): 689-701. DOI: 10.3856/vol49-issue5-fulltext-262. 

Ighwela KA, Ahmed AB, Abol-Munafi AB. Condition Factor as an Indicator of Growth and Feeding Intensity of Nile Tilapia Fingerlings (Oreochromis niloticus) Feed on Different Levels of Maltose. American-Eurasian J. Agric. & Environ. Sci. 2011; 11 (4): 559-563. 

Komba EA, Munubi RN, Chenyambuga SW. Comparison of body length-weight relationship and condition factor for Nile tilapia (Oreochromis niloticus) cultured in two different climatic conditions in Tanzania. International Journal of Fisheries and Aquatic Studies. 2020; 8 (3): 44-48. 

Shahabuddin AM, Khan MND, Ayna E, Wonkwon K, Murray WW, Yoshimatsu T, et al. Length-weight Relationship and Condition Factor of Juvenile Nile Tilapia Oreochromis niloticus (Linnaeus 1758) Fed Diets with Pyropia spheroplasts in Closed Recirculating System. Asian Fisheries Science, 2015; 28: 117-129.

P17 L496: Se-Np? Kindly use uniform terms in whole manuscript when representing selenium as nanoparticles like

L499: Se-Np?

L500: N-Se?

506: Se-Nps?

508: Se-Nps?

We apologize for the inconsistent use of such terminology. We have revised and consistently used Se-Nps throughout the manuscript.

P17 L514-516: how the present results compared with human Se requirement? I consider comparing results with other fish species.

We have narrated our results while comparing with relevant findings from otrher studies. 

P18 L545-549: the paragraph may be shifted to introduction section. Not needed here.

We have removed these sentences for better clarity.

P19 L556-562: again these sentences are not required here. The authors have to write significant benefits pertaining to fish.

Revised.

P19 l570-576: repetition of sentences needs revision.

Revised.

Conclusion: not sound needs revision.

Revised.

---

## [Decision Letter · Decision Letter 1]

28 Apr 2022

To 14/01/2022 The EditorEffects of selenium nanoparticle on the growth performance and nutritional quality in Nile Tilapia, Oreochromis niloticus

PONE-D-22-01541R1

Dear Dr. Das,

We’re pleased to inform you that your manuscript has been judged scientifically suitable for publication and will be formally accepted for publication once it meets all outstanding technical requirements.

Kind regards,

Mahmoud A.O. Dawood, PhD

Academic Editor

PLOS ONE

Additional Editor Comments (optional):

Reviewers' comments:

Reviewer's Responses to Questions

**Comments to the Author**

1. If the authors have adequately addressed your comments raised in a previous round of review and you feel that this manuscript is now acceptable for publication, you may indicate that here to bypass the “Comments to the Author” section, enter your conflict of interest statement in the “Confidential to Editor” section, and submit your "Accept" recommendation.

Reviewer #1: All comments have been addressed

Reviewer #2: All comments have been addressed

2. Is the manuscript technically sound, and do the data support the conclusions?

Reviewer #1: Yes

Reviewer #2: Yes

3. Has the statistical analysis been performed appropriately and rigorously? 

Reviewer #1: Yes

Reviewer #2: Yes

4. Have the authors made all data underlying the findings in their manuscript fully available?

Reviewer #1: Yes

Reviewer #2: Yes

5. Is the manuscript presented in an intelligible fashion and written in standard English?

Reviewer #1: Yes

Reviewer #2: Yes

6. Review Comments to the Author

Reviewer #1: The authors have addressed my comments and amended the text accordingly, so these findings can be accepted for publication.

Reviewer #2: The authors addressed most of the queries in the revised manuscript and is now acceptable for publication

7. PLOS authors have the option to publish the peer review history of their article (what does this mean?). If published, this will include your full peer review and any attached files.

Reviewer #1: No

Reviewer #2: **Yes: **Imtiaz Ahmed Associate Professor, Department of Zoology, University of Kashmir, India

---

## [Editor Report · Acceptance letter]

17 May 2022

PONE-D-22-01541R1 

Effects of selenium nanoparticle on the growth performance and nutritional quality in Nile Tilapia, Oreochromis niloticus 

Dear Dr. Das:

I'm pleased to inform you that your manuscript has been deemed suitable for publication in PLOS ONE. Congratulations! Your manuscript is now with our production department. 

Kind regards, 

on behalf of

Dr. Mahmoud A.O. Dawood 

Academic Editor

PLOS ONE